**Data Availability Statement:** The raw data cannot be publicly deposited because of privacy reasons. The questionnaires and templates for the

# Pharmacogenomics decision support in the U-PGx project: Results and advice from clinical implementation across seven European countries

Kathrin Blagec[1], Jesse J. Swen[2], Rudolf Koopmann[3,4], Ka-Chun Cheung[5], Mandy Crommentuijn - van Rhenen[5], Inge Holsappel[5], Lidija Konta[3], Simon Ott[1], Daniela Steinberger[3,4], Hong Xu[1], Erika Cecchin[6], Vita Dolžan[7], Cristina Lucía Dávila-Fajardo[8], George P. Patrinos[9], Gere Sunder-Plassmann[10], Richard M. Turner[11], Munir Pirmohamed[12], Henk-Jan Guchelaar[2], Matthias Samwald 📍[1]\*, Ubiquitous Pharmacogenomics Consortium[¶]

1 Institute of Artificial Intelligence, Center for Medical Statistics, Informatics, and Intelligent Systems, Medical University of Vienna, Vienna, Austria, 2 Department of Clinical Pharmacy & Toxicology, Leiden University Medical Center, Leiden, The Netherlands, 3 Diagnosticum Center for Human Genetics, Frankfurt am Main, Germany, 4 Institute for Human Genetics, Justus Liebig University, Giessen, Germany, 5 Medicines Information Centre, Royal Dutch Pharmacists Association (KNMP), The Hague, The Netherlands, 6 Experimental and Clinical Pharmacology Unit, Centro di Riferimento Oncologico di Aviano (CRO) IRCCS, Aviano, Italy, 7 Faculty of Medicine, Institute of Biochemistry and Molecular Genetics, Pharmacogenetics Laboratory, University of Ljubljana, Ljubljana, Slovenia, 8 Clinical Pharmacy Department, Hospital Universitario Virgen de las Nieves, Instituto de Investigación Biosanitaria Granada (Ibs.Granada), Granada, Spain, 9 Department of Pharmacy, Laboratory of Pharmacogenomics and Individualized Therapy, University of Patras School of Health Sciences, Patras, Greece, 10 Division of Nephrology and Dialysis, Department of Internal Medicine III, Medical University of Vienna, Vienna, Austria, 11 The Wolfson Centre for Personalised Medicine, Institute of Systems, Molecular and Integrative Biology, University of Liverpool, Liverpool, United Kingdom, 12 Department of Molecular and Clinical Pharmacology, Royal Liverpool University Hospital and University of Liverpool, Liverpool, United Kingdom

¶ Membership of the Bunny Genome Sequencing Consortium is provided in the Acknowledgments.
\* matthias.samwald@meduniwien.ac.at

## Abstract

### Background

The clinical implementation of pharmacogenomics (PGx) could be one of the first milestones towards realizing personalized medicine in routine care. However, its widespread adoption requires the availability of suitable clinical decision support (CDS) systems, which is often impeded by the fragmentation or absence of adequate health IT infrastructures. We report results of CDS implementation in the large-scale European research project Ubiquitous Pharmacogenomics (U-PGx), in which PGx CDS was rolled out and evaluated across more than 15 clinical sites in the Netherlands, Spain, Slovenia, Italy, Greece, United Kingdom and Austria, covering a wide variety of healthcare settings.

### Methods

We evaluated the CDS implementation process through qualitative and quantitative process indicators. Quantitative indicators included statistics on generated PGx reports, median

requirements analysis, the risk assessment and the user survey are available as supplementary material. Further, the complete results of the risk assessment are available as supplementary material. The raw results of the user survey are not publicly available due to privacy concerns because data contain potentially identifying or sensitive information in accordance with regulation of ethics board of the Medical University of Vienna. Requests for data access need to be processed by the MedUni Vienna Data Clearing House, Vienna, Austria. Contact information for the data clearing house is datenclearing@meduniwien.ac.at.

**Funding:** This work was supported by European Community's Horizon 2020 Programme grant number 668353 (U-PGx), awarded to JJS, KCC, DS, EC, VD, CLDF, GPP, GSP, MP, HJG and MS. Funder did not play any role in study design, data collection and analysis, decision to publish, or preparation of the manuscript.

**Competing interests:** DS has developed concepts of genetic information management (GIM) that are realized by the company bio.logis digital health GmbH. For the company, she is CEO. She is also medical director of diagnosticum Center of Human Genetics, a diagnostic institution that is applying and testing the GIMsystems in the context of medical care. GPP is Full Member and National representative of the European Medicines Agency, Committee for Human Medicinal Products (CHMP) - Pharmacogenomics Working Party, Amsterdam, the Netherlands. This does not alter our adherence to PLOS ONE policies on sharing data and materials. The other authors have no competing interests to declare.

**Abbreviations:** AA, Automatic alerts; CDS, Clinical decision support; DPWG, Dutch Pharmacogenetics Working Group; DR, Digital report; EHR, Electronic health record; GIMS, Genetic information management system; LIMS, Laboratory information management system; PGx, Pharmacogenomics; MDR, Medical device regulation; PR, Paper-based report; SC, Safety Code card; U-PGx, Ubiquitous Pharmacogenomics.

time from sampled upload until report delivery and statistics on report retrievals via the mobile-based CDS tool. Adoption of different CDS tools, uptake and usability were further investigated through a user survey among healthcare providers. Results of a risk assessment conducted prior to the implementation process were retrospectively analyzed and compared to actual encountered difficulties and their impact.

## Results

As of March 2021, personalized PGx reports were produced from 6884 genotyped samples with a median delivery time of twenty minutes. Out of 131 invited healthcare providers, 65 completed the questionnaire (response rate: 49.6%). Overall satisfaction rates with the different CDS tools varied between 63.6% and 85.2% per tool. Delays in implementation were caused by challenges including institutional factors and complexities in the development of required tools and reference data resources, such as genotype-phenotype mappings.

## Conclusions

We demonstrated the feasibility of implementing a standardized PGx decision support solution in a multinational, multi-language and multi-center setting. Remaining challenges for future wide-scale roll-out include the harmonization of existing PGx information in guidelines and drug labels, the need for strategies to lower the barrier of PGx CDS adoption for healthcare institutions and providers, and easier compliance with regulatory and legal frameworks.

## Background

With the availability of guidelines that provide evidence-based therapeutic recommendations for more than 100 gene-drug pairs, clinical pharmacogenomics (PGx) has the potential to be one of the first milestones on the way to realize personalized medicine in routine clinical care [1–3]. Ever since the first PGx implementation programs were devised, clinical decision support (CDS) systems have proven to be indispensable components of PGx implementation by distilling the vast amount of information contained within a patient's PGx results into clinically actionable and concise therapeutic recommendations [4–8].

While the implementation of PGx-guided therapy has gained some traction in certain specialized early adopter sites, particularly in the US, within the past two decades, broader adoption has not yet occurred [5–7, 9]. General challenges to incorporation of PGx into clinical practice include lack of awareness and education about PGx among healthcare providers, lack of data on cost-effectiveness of multi-panel testing approaches and consequently reimbursement issues [10]. These challenges are further aggravated by the limitations and diversity of the current European infrastructural healthcare landscapes in terms of availability, capability and interconnectedness of Electronic Health Record Systems (EHRs), which form the prerequisite for the most efficient delivery of PGx CDS, i.e., through the display of automatic alerts during the prescribing process [11–14].

To address these challenges, the Ubiquitous Pharmacogenomics (U-PGx) consortium was initiated in 2016 with three primary aims: First, to address current barriers for PGx implementation including the provision of enabling tools. Second, to yield scientific evidence for the improvement of patient outcomes through pre-emptive PGx testing by means of a prospective,

randomized clinical study. And third, to evaluate the cost-effectiveness of such an approach [15, 16].

The project implements a so-called 'pre-emptive' PGx testing approach, where an individual is tested for a panel of multiple genes at once instead of conducting multiple single gene tests over time, facilitating PGx-guided treatment for a number of gene-drug combinations throughout an individual's life [5, 17–19].

To enable the clinical implementation of pre-emptive PGx testing across all seven participating European countries, despite large diversity in infrastructural conditions, the U-PGx consortium has conceptualized a unique multi-modal, multi-language PGx CDS framework that offers a standardized implementation framework while also meeting the individual technical requirements and capabilities of each clinical site [14].

The effectiveness of CDS hinges on its adoption by healthcare providers. Previous research has reported mixed results regarding satisfaction with and adoption of PGx CDS [20–24]. Further, the majority of previous PGx CDS evaluation studies was conducted in simulated study settings rather than in real-world clinical settings [19, 21, 23–27].

In contrast, this study investigated the implementation and usability of a multi-modal CDS infrastructure in a real-world, multinational clinical setting. Further, to the best of our knowledge, this is the first study that examines a CDS infrastructure that consists of multiple, complementary tools conceptualized to accommodate a variety of health IT infrastructures and capabilities. In particular, we also investigated the acceptance and usefulness of a mobile-based CDS tool that was designed as an auxiliary tool to enable using and sharing of PGx test results in healthcare settings where the EHR infrastructure is fragmented or missing altogether.

To evaluate the implementation process and the usability of the provided CDS tools, we used a triangulated research approach combining quantitative and qualitative data including a user study.

In the following, we describe process indicators on uptake and usability, an analysis of qualitative feedback on workflow efficiency, bottlenecks and optimization potential of the U-PGx implementation framework, and share insights and experiences from all phases and layers of the implementation process. Furthermore, we provide considerations on prerequisites and challenges for continuation and expansion of the PGx CDS implementation in Europe, including practical advice that can be of value for similar projects. For reporting our implementation strategy, we follow the Standards for Reporting Implementation Studies (StaRI) checklist (see S5 File) [28].

## Methods

### Setting

The U-PGx project is a large-scale, multinational European implementation project to investigate the impact of pre-emptive PGx testing on patient outcomes in a clinical study ("*PRE-PARE*", ClinicalTrials.gov Identifier: NCT03093818), and the cost-effectiveness of such an approach [15].

*PREPARE* was designed as a prospective, block-randomized, controlled study. More than 6,800 patients were enrolled over the course of three years at one or more clinical sites across the seven participating countries. The participating countries were randomized to start with either PGx-guided treatment or with 'standard of care' (control arm) in the first block of 18 months. After 18 months, the countries switched to the opposite arm and recruited new patients for another 18 months. The primary outcome measure was the occurrence of at least one clinically relevant adverse drug reaction caused by the drug of inclusion. Additional details on the study design can be found in [15].

More than 15 clinical sites across seven participating European countries (i.e., the Netherlands, Spain, Slovenia, Italy, Greece, United Kingdom and Austria) with varying health IT infrastructures—ranging from complete absence of any such infrastructure to the capability to display CDS alerts via the electronic health record—are participating in the project.

To address the complex requirements of the U-PGx project setup, a unique assortment of complementary multi-modal decision support tools was developed around a central knowledge management platform and successively implemented from 2017.

Pseudonymized raw results from the genotyping platform are automatically or manually transferred to the secure central platform, and patient-specific results and recommendations based on the guidelines authored by the Dutch Pharmacogenetics Working Group of the Royal Dutch Pharmacists Association for more than 40 drugs associated with one or more of 13 different genes can be retrieved in various formats and in seven different languages.

Depending on their local infrastructure and preferences, clinical sites chose to use one or more of the following CDS tools: 1) Semi-structured data that can be integrated with existing local EHRs and used to display automatic alerts in the prescribing process, 2) a conventional report format that can be printed or filed in the EHR, and 3) a credit-card sized "Safety Code" card (www.safety-code.org) that is handed to the patient and enables on-the-fly access of patient-specific dosing recommendations via a smartphone or tablet using QR codes (Figs 1

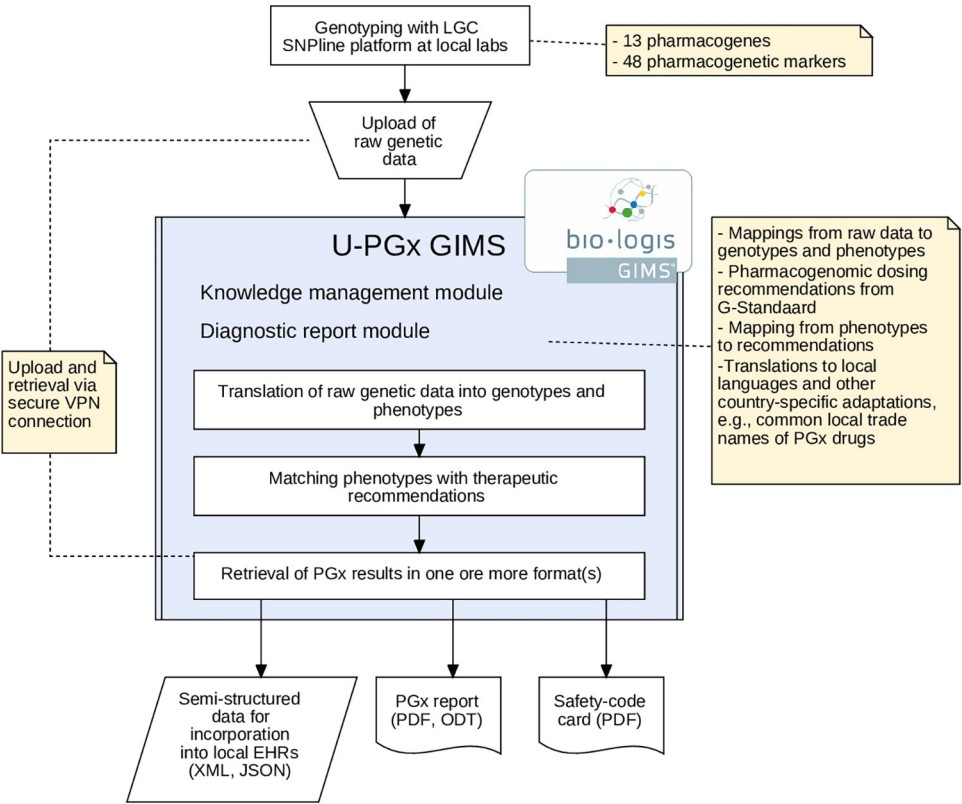

**Fig 1. Framework of complementary decision support solutions implemented in the U-PGx project.** The provided set of solutions accommodates the needs and requirements of the highly diverse implementation sides while guaranteeing a standardized intervention. Clinical sites with the infrastructural capabilities for interruptive CDS alerts in the form of pop-up messages in the EHR or e-prescription system are provided with semi-structured data. For all other sites, passive CDS can be delivered either inside the EHR system as a digital report, or outside the EHR system via mobile- and paper-based solutions. A more detailed technical description is available in [14].

and 2). The technical set-up and implementation of these tools, and the IT infrastructures and capabilities of each participating clinical site have been described in detail previously [14].

## Pre-implementation phase

Fig 3 shows the different phases of the implementation process including the tools and methods used for monitoring and evaluation. The different phases including evaluation methods are described in detail in the following sections.

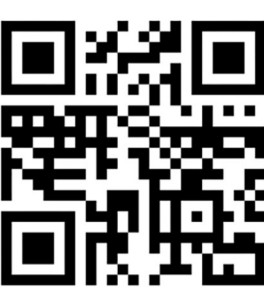

**safety-code**
*The **Medication Safety Code** initiative*

I participate in the U-PGx *PREPARE study* (study arm). For more information, please visit www.upgx.eu/study

To the healthcare provider:
Please scan the QR code to view pharmacogenomics-based drug dosing recommendations for this patient.

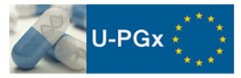

**Contact**
Lab name
E-mail address
Phone
Additional contact information

**safety-code**
*The **Medication Safety Code** initiative*

Name: Jane Doe
Date of birth: 01.02.1934

| *Gene, status* | *Critical drug substances (modification recommended!)* |
|---|---|
| *CYP2C19*<br>Poor metabolizer | Clopidogrel, Sertraline |
| *CYP2D6*<br>Ultrarapid metabolizer | Amitriptyline, Clomipramine, Codeine, Doxepin, Haloperidol, Imipramine, Metoprolol, Nortriptyline, Propafenone, Tramadol, Venlafaxine |
| *TPMT*<br>Poor metabolizer | Azathioprine, Mercaptopurine, Thioguanine |
| Other tested genes<br>Not actionable | *CYP2B6, CYP2C9, CYP3A5, DPYD, F5, HLA-A3101, HLA-B1502, HLA-B5701, SLCO1B1, TPMT, UGT1A1, VKORC1* |

Date printed: 15.06.2017

**Fig 2. Front and back side of an exemplary Safety Code card for a fictional patient.** The Safety Code card is given to all patients recruited for the 'PGx-guided treatment' arm of the study. Patients recruited for the control arm were eligible to receive a Safety Code card at the end of the study [14].

## Pre-implementation

### Analysis of European regulatory and legal conditions

### Requirements analysis: Initial structured questionnaire and consequent individual follow-up

### Risk assessment: Structured questionnaire

## Post-implementation: Monitoring and evaluation

**Quantitative process indicators:** Quality control and statistics on report generation, Dashboard for monitoring report retrievals via Safety Code card

**Qualitative process indicators:** Web-based evaluation survey among healthcare providers, Structured questionnaire on workflow efficiency and options for improvement among clinical coordinators

## Maintenance

**Analysis of needs and pre-requisites:** Structured questionnaire, Preparation of a workshop

## Communication and project management tools

**Mailing lists:** Technical support, Communication with clinical teams

**Online survey platform:** User survey, Risk assessment

**Gantt Chart Dashboard:** Timeline and implementation status tracking

**Fig 3. Phases and tools of the implementation process.**

**Requirements analysis.**   One of the main challenges of a large-scale implementation project covering many diverse institutions and geographic regions is to provide a solution that is standardized yet flexible enough to accommodate different existing frameworks and conditions. Requirements at the different sites were assessed in an iterative process.

To gain a first overview of the current infrastructure and workflows in place at the participating institutions, a structured questionnaire was developed together with the coordinators

and collaborators involved in developing and implementing the IT solutions. Covered topics included: 1) if, available, the current workflow for selecting and testing patients, 2) the storage of test results and interpreted PGx data, and 3) the local EHR infrastructure and capabilities. The complete requirements analysis template and results are provided in S1 File.

After the initial assessment, requirements of each implementation site were further specified individually. Additionally, an analysis of legal and regulatory conditions relevant to the implementation was conducted (see S2 File).

**Risk assessment.** A list of potential risks was compiled based on challenges and reasons for delay reported by other decision support implementation projects, the information collected in the requirements analysis and the analysis of European regulatory frameworks [4]. Based on these sources, a questionnaire was drafted, in which both clinical collaborators and all partners involved in the implementation were asked to rate the listed risks in terms of 1) expected probability (low, medium, high) and 2) expected impact (minor, medium, severe) of the respective risk (see S3 File).

The severity of identified risks was estimated based on their impact and probability level using a 3x3 risk assessment matrix. Risks with high impact and medium or high probability were rated as "severe". Risks with low probability and high impact, and risks with medium impact and medium or high severity were rated as "medium" [29, 30].

## Post-implementation phase

**Evaluation framework.** To monitor and gain a comprehensive view of the uptake and acceptance of the decision support tools, various quantitative and qualitative data collection methods were combined and triangulated. These methods included 1) tracking of general process indicators, 2) a standardized questionnaire targeted at the clinicians and pharmacists involved in the recruiting or treatment of patients enrolled in the PREPARE study, 3) statistics on report retrievals via the Safety Code card component of the decision support tools, and 4) qualitative feedback by the clinical study coordinators.

**General process indicators.** The centralized processing of samples and generation of PGx reports using the genetic information management system (GIMS) allowed the tracking of process indicators across all sites (see Fig 1). These included the number of uploaded samples and generated reports per implementation site, the turnaround time between sample upload and report delivery and genotype/phenotype frequencies. Data on process indicators was extracted from GIMS, and descriptive statistics were calculated.

Due to the design of the clinical study as a randomized crossover study, roll-out of the CDS infrastructure was scheduled in two phases, with the first group of implementation sites being scheduled for the beginning of 2017, and the second group of sites for July 2018. Data on planned roll-out dates vs. actual roll-out dates of the CDS infrastructure including reasons for delays were collected for each implementation site.

**Survey.** Informed by previous PGx CDS user studies, a questionnaire to evaluate the decision support tools was developed combining quantitative and qualitative question formats [21, 25]. Queried information included (1) demographic data, (2) the self-reported extent of adoption of PGx testing and the various CDS tools, and (3) the usability of the various CDS tools in terms of their perceived information content, usefulness and workflow integrability.

The questionnaire was implemented as a web-based survey using the online survey platform "Alchemer" and reviewed and tested within the consortium [31]. The questionnaire is available in S4 File.

Survey participants were identified through communication with the principal clinical investigators at the respective study sites and invited via email to participate in the online web-

based survey. Response status was tracked and three reminder invitations were sent in case of no response.

Data analysis was conducted using the Python programming language and descriptive statistics were calculated using the Python NumPy and pandas libraries [32, 33]. Plots to visualize Likert items were created using Python matplotlib [34].

**Statistics on report retrievals.** The utilization of the Safety Code card component of the clinical decision support tools was evaluated by collecting de-identified statistics on page retrievals via the card's QR code. A dashboard to monitor results was created using the Python programming language.

**Qualitative feedback by implementation site coordinators.** In addition to the user survey and quantitative process indicators, the workflow and implementation were qualitatively evaluated by surveying the clinical implementation site coordinators on the implemented workflow, encountered workflow bottlenecks (i.e., components or processes that negatively impact report delivery time), their satisfaction with the different components of the decision support infrastructure and their interest in and prerequisites for continuing to use the different components.

### Ethics approval and consent to participate

This study received ethics approval from the ethics committee of the Medical University of Vienna (No. 2091/2016). All participants provided informed consent. Furthermore, the study was approved by ethics review boards in all participating countries and all necessary permits and approvals were obtained. The review boards in other participating countries were: 1) Comité Coordinador de Ética de la Investigación Biomédica de Andalucía, Sevilla (Spain), 2) Scientific Council of the Committee on research and ethics, University General Patras Hospital (Greece), 3) Comitato etico unico regionale, Istituto di ricovero e cura a carattere scientifico, Aviano (Italy), 4) Medisch Ethische Toetsingscommissie Leiden Den Haag Delft (Netherlands), 5) Komisiji za medicinsko etiko, University of Ljubljana (Slovenia), 6) NHS Health Research Authority, North West—Liverpool Central Research Ethics Committee (UK).

## Results

### Risk assessment

Overall, 27 risks were identified of which seven (25.9%) were rated as severe (i.e., having a severe impact and having medium or high probability). 12 (44.4%) risks were rated as medium and 8 (29.6%) as minor. Out of the seven risks rated as severe, two were related to team coordination and communication while four were related to IT and lab infrastructure issues (see Table 1).The complete risk assessment results are available in S3 File.

Risks assigned the highest severity rating were collaboration and communication challenges resulting from the large-scale, multidisciplinary setting of the project, establishing collaboration including obtaining necessary permissions from local IT departments, risks associated with establishing the data basis for the CDS solutions and devising a workflow and infrastructure suitable for each clinical site.

This structured, collaborative assessment of risks was then used to categorize risks into different action categories, i.e., 'avoid/control/transfer' and 'watch/assume' and to devise mitigation strategies, where possible. In retrospect, risks that emerged as the greatest challenges were institutional, collaborative and bureaucratic factors, such as dependency on collaborators and suppliers outside of the project for whom the project was not a priority, and the development of the required tools and reference data resources (see Section 'General implementation process indicators').

**Table 1. Identified risks per thematic category and severity rating.** LIMS: Laboratory information management system.

| Category | Severe (avoid, control, transfer) | Medium (avoid, control, transfer) | Minor (watch, assume) |
|---|---|---|---|
| **Team coordination and communication** | Collaboration and availability of different suppliers participating in the project | Close collaboration between ICT project partners and local sites and ICT departments required (n = 2) | N/A |
| | Lack of prioritization of essential tasks due to institutional factors | Financial compensation may be required for collaboration with local ICT departments | |
| **IT / Lab infrastructure** | Requirement of approval of local institutional committees | Required to remotely work with a large number of differing local IT infrastructures, LIMS and EHRs and their interfaces (n = 2) | Laboratory that will provide the PGx results will move to another building during the project |
| | Local requirements related to data format (e.g., XML) | Lack of EHR/LIMS in some countries | Quality and consistency of translations of guidelines to local languages need to be guaranteed |
| | Highly diverse requirements for interfaces that may require customized solutions | Generation of paper-based reports required in addition to solutions for EHR systems | |
| | Difficulties in implementing rules for decision support | Infrastructure and workflow for printing safety-code card needs to be devised | |
| | | Methods for de-identification required | |
| | | Training in ICT-related topics required for clinical partners | |
| **Staffing** | - | Problems with hiring personnel due to bureaucratic factors (n = 2) | Difficulties in getting staff with appropriate expertise (n = 2) |
| | | Extensive training required for new staff | |
| **Changes at implementation site** | N/A | N/A | Moving between hospital buildings will occur during the project (n = 2) |
| | | | Additional departments will likely be required to be involved in the project |
| **Other problems** | Resulting report based on DPWG guidelines too long or too unstructured to be usable by clinicians | N/A | N/A |
| **Total number** | **7** | **13** | **7** |

## General implementation process indicators

For the first group, roll-out was finalized in June 2017, and for the second group in October 2018, resulting in postponements of six and three months, respectively. In the first case, delays were primarily caused by challenges including institutional factors and development of the required tools and reference data resources, such as the translation tables for conversion of variants to haplotypes and genotype-predicted phenotypes; the latter entailed many complexities due to missing standardized nomenclatures. For the second group, the recorded delay was attributable to the postponement of the first phase.

As of March 2021, PGx reports and Safety Code cards were created for 6884 genotyped samples in U-PGx GIMS. Processing of samples of control arm patients is still ongoing at the time of writing. Fig 4 shows the number of uploaded samples per country. For 94.5% of samples, a full report covering all SNPs of the U-PGx genotyping panel could be delivered. For the remaining samples, a partial report was delivered due to missing results for one or more SNPs.

As of March 2021, 83.8% of uploaded samples had at least one actionable phenotype, i.e., a phenotype for which the DPWG guidelines recommended an adjustment of treatment. The median number of actionable phenotypes per sample was one. Median waiting time for report generation after sample upload was twenty minutes.

## Decision support infrastructure evaluation survey

**Demographics.** In total, 131 healthcare providers involved in recruiting and treating patients enrolled in the PREPARE study were invited to participate in the survey. Out of these,

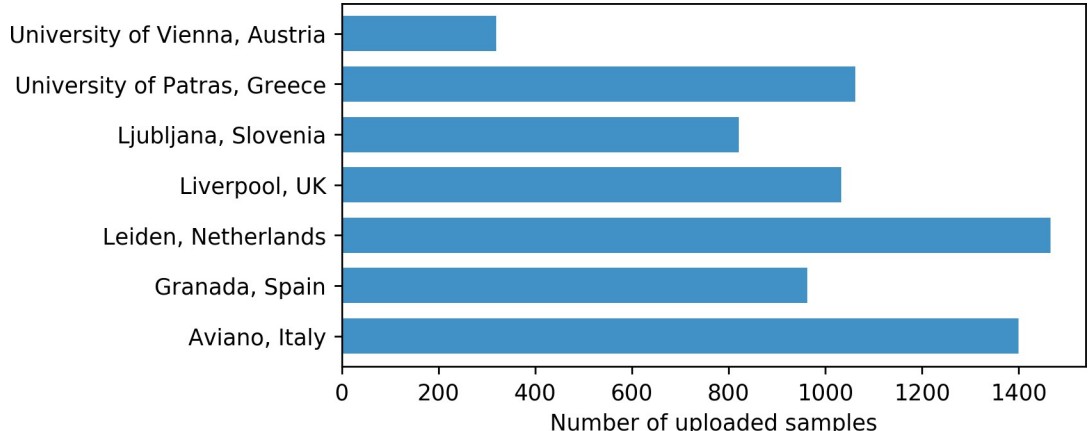

**Fig 4. Number of genotyping samples uploaded per country as of March 2021.** Processing of samples of control arm patients was still ongoing at the time of writing.

65 completed the questionnaire (Response rate 49,6%). Fig 5 shows the number of respondents vs. the number of invited healthcare providers per country and the distribution of respondents across countries.

Forty percent of participants in our sample identified as male, 58.5% as female and one participant preferred not to state their gender. This ratio corresponded approximately to the ratio among the total number of invited healthcare providers (59.5% female). Participants had a median age of 40 years, and the median number of years of working experience was twelve. The median number of years of experience with PGx was two (see Table 2).

The majority of respondents (61.5%) stated to be mostly treating patients in an outpatient setting. Pharmacy (30.8%), primary care / family medicine (15.4%) and psychiatry (16.9%) were the three most often reported fields of work, followed by oncology (23.1%), cardiology (9.2%) and nephrology (4.6%). Corresponding percentages for all invited users based on the type of healthcare facilities participants were recruited from were pharmacy (18.3%), primary care / family medicine (27.5%), psychiatry (8.4%), oncology (29.0%), cardiology (11.5%) and nephrology (5.3%).

The majority of participants agreed or strongly agreed to be comfortable using computers, mobile devices and EHRs (Fig 6). Out of these three devices, EHRs and computers had the

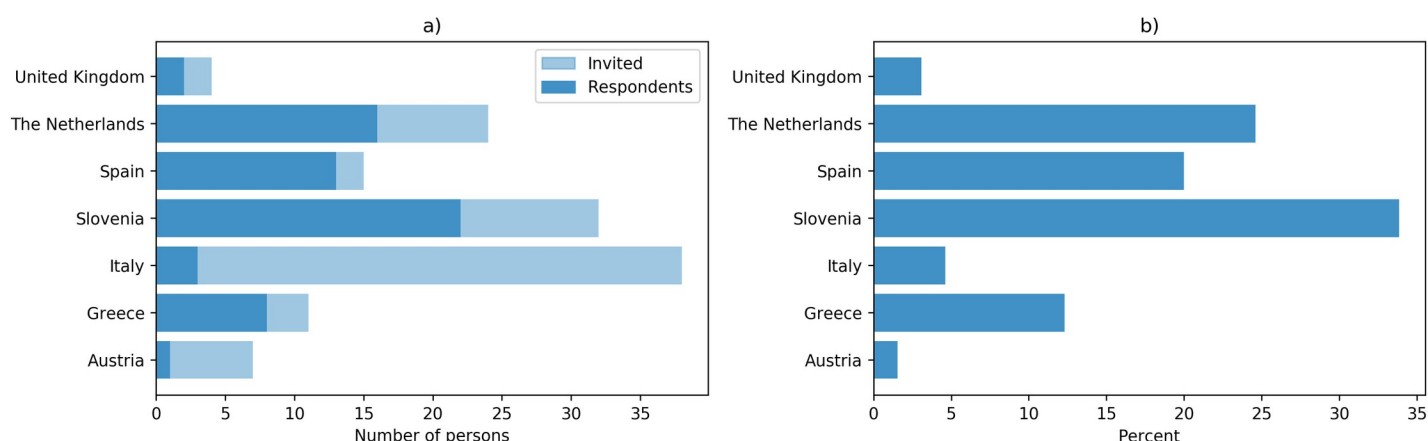

**Fig 5.** a) Invited healthcare providers vs. number of respondents per country, b) Percentage of respondents per country.

**Table 2. Demographics of the survey participants.**

|  | Median | Q₁ | Q₃ | Max | Min |
|---|---|---|---|---|---|
| Age | 40 | 35 | 50 | 62 | 22 |
| Years of working experience | 12.0 | 8 | 20 | 36.0 | 0.5 |
| Years of experience with PGx | 2.0 | 0.5 | 2 | 12.0 | 0.1 |
| Number of PGx tests ordered per week | 1 | 0 | 1 | 9.0 | 0.0 |

highest number of participants stating to be not comfortable using them (both n = 5, 7.7%) or being neutral towards using them (n = 3, 4.6% and n = 10, 15.4%, respectively). Respondents who indicated unfamiliarity or being uncomfortable with one or more of these devices were from Slovenia or Greece, i.e. countries with no or only partially available EHR infrastructure.

When asked whether they had received enough training in PGx, more than two thirds of respondents agreed (n = 39, 60%) or strongly agreed (n = 6, 9.2%), while 17 (26.2%) participants felt undecided and three felt they had received too little training.

**Uptake and usage.** Table 3 shows the number of users and reported median times of use per week for the different CDS tools.

Most participants had at least once used the paper-based reports (76.9%) or digital reports (41.5%). 23% of respondents had used automatic alerts via the EHR and 33.8% of respondents indicated to have at least once used the Safety Code card.

In four cases, participants reported to have used other methods to interpret a patient's PGx results than the provided CDS tools. These included direct communication or communication via phone calls, e-mail or WhatsApp messages with pharmacists.

**Usability evaluation.** Across all tools, the majority of users felt that they integrate well with their work routine (Fig 7). The paper-based reports had the highest number of users who

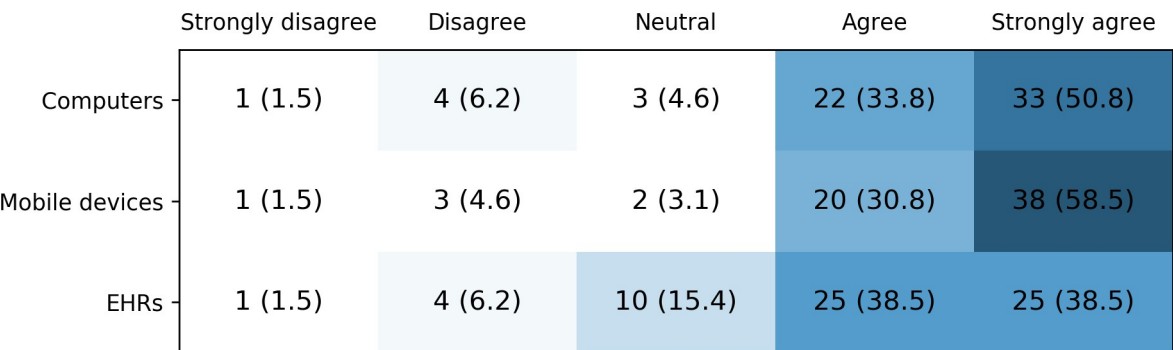

### I am comfortable with computers, mobile devices and EHRs

|  | Strongly disagree | Disagree | Neutral | Agree | Strongly agree |
|---|---|---|---|---|---|
| Computers | 1 (1.5) | 4 (6.2) | 3 (4.6) | 22 (33.8) | 33 (50.8) |
| Mobile devices | 1 (1.5) | 3 (4.6) | 2 (3.1) | 20 (30.8) | 38 (58.5) |
| EHRs | 1 (1.5) | 4 (6.2) | 10 (15.4) | 25 (38.5) | 25 (38.5) |

**Fig 6. Healthcare provider's self-reported comfortableness with computers, mobile devices and electronic health records (EHRs).**

**Table 3. Reported usage of the different CDS tools among survey participants.** More than one CDS tool can be used by a single user.

|  | Number of users | Median number of uses per week | Max number of uses per week | Min number of uses per week |
|---|---|---|---|---|
| Automatic alerts | 15 | 1 | 5 | 0 |
| Digital reports | 27 | 2 | 10 | 0 |
| Paper-based reports | 50 | 1 | 30 | 0 |
| Safety Code cards | 22 | 1 | 5 | 0 |
| Other methods | 4 | 1 | 1 | 1 |

## The CDS tool integrates well with my work routine

| | Strongly disagree | Disagree | Neutral | Agree | Strongly agree |
|---|---|---|---|---|---|
| SC | 1 (1.5) | 2 (3.1) | 5 (7.7) | 14 (21.5) | 0 (0.0) |
| PR | 0 (0.0) | 1 (1.5) | 11 (16.9) | 31 (47.7) | 7 (10.8) |
| DR | 0 (0.0) | 1 (1.5) | 5 (7.7) | 16 (24.6) | 5 (7.7) |
| AA | 0 (0.0) | 0 (0.0) | 1 (1.5) | 10 (15.4) | 4 (6.2) |

**Fig 7. Healthcare providers' rating of workflow integration of the different CDS tools.** SC: Safety Code cards, PR: printed reports, DR: digital reports, AA: automatic alerts.

felt neutral about this (n = 11), followed by the digital reports (n = 5) and the Safety Code card (n = 5). For these tools, only few users indicated that they do not fit into their work routine (n = 5 in total).

Similarly, across all tools, the majority of users felt that the tools provided the right amount of information (Fig 8). However, for all tools, there were also single users who either felt that too little or too much information was provided. Feeling that too much information is provided was most salient for the Spanish implementation site, where a user also pointed out a preference for their own institutional PGx reports over the ones delivered as part of the project. Free text comments about the CDS information provided indicated the wish for more concrete information for whether action needs to be taken and the wish for more practical recommendations. Many users felt that both the paper-based reports and the digital reports were generally too long, should include less information and could be improved by better structuring, which was especially salient for the paper-based reports.

Most respondents agreed or strongly agreed to have received enough training to confidently use the different CDS tools (Fig 9). Again, several users, mostly from Greece and the Netherlands, also felt undecided and few felt they had received too little training.

## Do you feel the tool provides you with too much, too little, or just the right amount of information for use in clinical practice?

| | Too little information | Just the right amount of information | Too much information |
|---|---|---|---|
| SC | 4 (18.2) | 15 (68.2) | 3 (13.6) |
| PR | 3 (6.0) | 42 (84.0) | 5 (10.0) |
| DR | 1 (3.7) | 18 (66.7) | 4 (14.8) |
| AA | 3 (20.0) | 9 (60.0) | 1 (6.7) |

**Fig 8. Healthcare providers' perception of the information amount of the different CDS tools.** SC: Safety Code cards, PR: printed reports, DR: digital reports, AA: automatic alerts.

I feel that I have received enough training to confidently use the CDS tool.

| | Strongly disagree | Disagree | Neutral | Agree | Strongly agree |
|---|---|---|---|---|---|
| SC | 0 (0.0) | 2 (9.1) | 8 (36.4) | 12 (54.5) | 0 (0.0) |
| PR | 0 (0.0) | 4 (8.0) | 10 (20.0) | 32 (64.0) | 4 (8.0) |
| DR | 0 (0.0) | 1 (3.7) | 6 (22.2) | 14 (51.9) | 6 (22.2) |
| AA | 0 (0.0) | 1 (6.7) | 3 (20.0) | 8 (53.3) | 3 (20.0) |

**Fig 9. Healthcare providers' perceived sufficiency of the received training on the different CDS tools.** SC: Safety Code cards, PR: printed reports, DR: digital reports, AA: automatic alerts.

For all tools, the user-friendliness was rated as "good" or "excellent" by the majority of users (Fig 10). Few users, most pronounced for the Spanish implementation site, rated the user friendliness as "Poor" with the paper-based reports having the highest number of this rating. The other categories "Worst imaginable", "Awful" and "Best imaginable" were not chosen by any respondents.

For all tools, the majority of users stated to be either "satisfied" or "very satisfied" with them (Fig 11). The paper-based reports and Safety Code card had the highest number of users who felt neither satisfied nor dissatisfied with them. Only two users indicated to be dissatisfied or very dissatisfied with a tool (i.e. the digital report and the Safety Code card). In the free text comments, the Safety Code card, however, received several positive comments. One user stated:

> "*A useful tool for which a caregiver can assume patients need to have it with [them] "all the time" Limited information can be printed, obviously. But a useful code (QR) can help to get more information about [the] genetic profile*"

## Report retrievals via the Safety Code card

In total, 4,040 report retrievals via the Safety Code card were logged from April 2018 until the end of patient recruitment on June 30 2020 (see Fig 12). Logging of report retrievals started in

Overall, I would rate the user-friendliness of the tool as

| | Worst imaginable | Awful | Poor | OK | Good | Excellent | Best imaginable |
|---|---|---|---|---|---|---|---|
| SC | 0 (0.0) | 0 (0.0) | 1 (4.5) | 8 (36.4) | 7 (31.8) | 6 (27.3) | 0 (0.0) |
| PR | 0 (0.0) | 0 (0.0) | 4 (8.0) | 15 (30.0) | 22 (44.0) | 9 (18.0) | 0 (0.0) |
| DR | 0 (0.0) | 0 (0.0) | 2 (7.4) | 5 (18.5) | 11 (40.7) | 9 (33.3) | 0 (0.0) |
| AA | 0 (0.0) | 0 (0.0) | 0 (0.0) | 5 (33.3) | 5 (33.3) | 5 (33.3) | 0 (0.0) |

**Fig 10. Healthcare providers' rating of the user-friendliness of the different CDS tools.** SC: Safety Code cards, PR: printed reports, DR: digital reports, AA: automatic alerts.

## Overall, how satisfied or dissatisfied are you with the tool?

| | Very Dissatisfied | Dissatisfied | Neutral | Satisfied | Very Satisfied |
|---|---|---|---|---|---|
| SC | 1 (4.5) | 0 (0.0) | 7 (31.8) | 11 (50.0) | 3 (13.6) |
| PR | 0 (0.0) | 0 (0.0) | 10 (20.0) | 36 (72.0) | 4 (8.0) |
| DR | 0 (0.0) | 1 (3.7) | 3 (11.1) | 21 (77.8) | 2 (7.4) |
| AA | 0 (0.0) | 0 (0.0) | 3 (20.0) | 8 (53.3) | 4 (26.7) |

**Fig 11. Healthcare providers' overall satisfaction with the different CDS tools.** SC: Safety Code cards, PR: printed reports, DR: digital reports, AA: automatic alerts.

April 2018, thus retrievals before this date are not included. Since the end of recruitment until the time of writing (beginning of March 2021), another 1,282 report retrievals were logged. With regard to this, it is important to note that patients recruited for the control arm only received their Safety Code cards after completion of the study.

In relation to the number of recruited patients, the Netherlands and Austria had the highest number of report retrievals, while Italy and Greece had the lowest. Returning visits (i.e. the same user retrieving more than one report) made up 61.6% of all report retrievals. Fig 12 shows the distribution of report retrievals over time.

### Workflow analysis including qualitative feedback by implementation site coordinators

Overall, the prepared workflow ranging from identification of patients for PGx testing to result retrieval was followed by all implementation sites with only minor divergences, and was generally deemed efficient by clinical coordinators. Exceptions included unexpected longer time spans from sample retrieval to report submission caused by issues concerning the genotyping platform that were experienced by several sites. Suggestions for improvement included a more transparent and user-friendly workflow for identifying, translating and annotating the updated PGx recommendations in the GIMS platform.

Both Spanish and Greek implementation sites reported a low usage of the Safety Code card, because of low acceptance from the patients' side, the healthcare providers' side or both.

All sites expressed interest in continuing several or all components of the CDS infrastructure in the future provided a continued updating of the guidelines and maintenance of the related CDS tools.

### Discussion

In this paper, we presented the results of the first European large-scale, multi-center, multi-national implementation of PGx decision support using a triangulated evaluation strategy. We combined quantitative and qualitative process indicators with qualitative meta-feedback to gain a comprehensive picture of the overall success of our implementation strategy and to identify areas for improvement, as well as remaining challenges to an extended large-scale roll-out of PGx decision support in Europe.

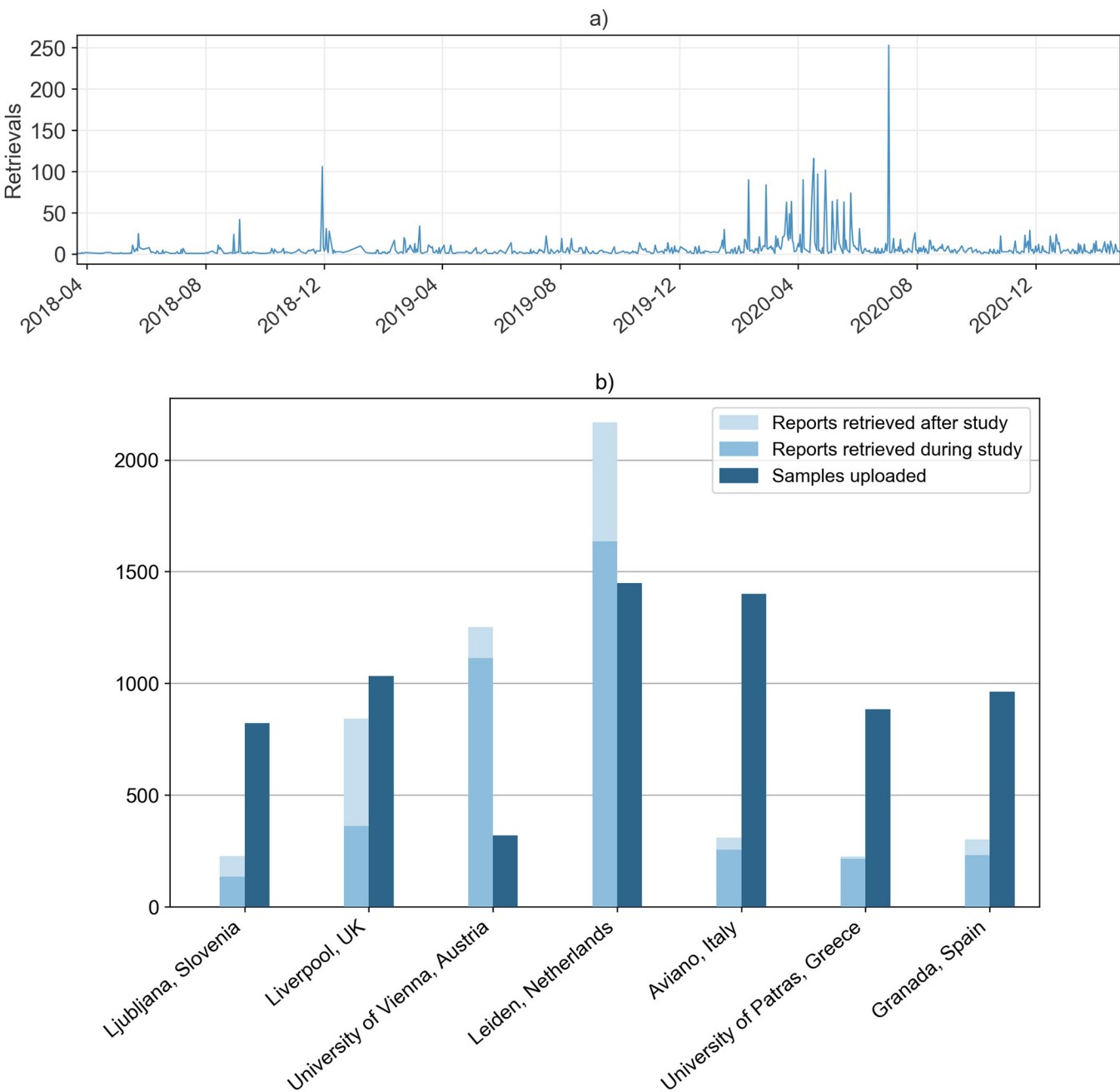

**Fig 12.** a) Report retrievals from April 2018 to March 2021. Report retrieval logging started in April 2018, thus retrievals before this date are not included. b) Samples uploaded vs. number of report retrievals via the Safety Code card during and after the study per implementation site. Patients recruited for the control arm received their Safety Code cards only after completion of the study.

## Evaluation of the implementation

Overall, the acceptance and satisfaction with the different CDS tools was rated positively across all investigated aspects (i.e. workflow integration, user-friendliness, information content, overall satisfaction) by the majority of involved healthcare providers. The majority of users further indicated that they felt they had both received enough training to confidently use the CDS tools and to use PGx results in general. In contrast, previous CDS implementation projects

have reported mixed results on the satisfaction with PGx CDS and general training in PGx [20–24, 35–37].

There are several possible explanations for the largely positive evaluation results of the U-PGx CDS infrastructure. Most importantly, the U-PGx CDS implementation strategy differentiates from previous projects in that it offers implementation sites a spectrum of tools that can be chosen and combined based on infrastructural requirements and subjective preference. Moreover, the implementation was accompanied by a dedicated education program, offering workshops and an e-learning program to further educate and familiarize participating healthcare providers with all aspects of PGx. This education program was informed by a survey that assessed healthcare provider's current knowledge about PGx at the beginning of the project [36, 38]. Further, a 'train-the-trainer' concept was used to increase the range of educational efforts, which means that project coordinators at each implementation site were trained and instructed to in turn train on-site personnel. Finally, it has to be noted that our sample of healthcare providers had, in median, two years of experience with at least some aspect of PGx, which may have made it easier for them to use the CDS tools.

While these user study results indicate an overall successful implementation, especially the opinions of less satisfied users may help to identify leverage points to increase overall acceptance among healthcare providers and future extended implementations outside the setting of a clinical study.

Areas for improvement that emerged from our evaluation included the structure and formatting of reports, and the phrasing and content of recommendations. It is well known from previous research that healthcare providers prefer PGx CDS recommendations to be concise with additional information (including links to brief PGx educational pieces) being available on request to avoid information overload [21, 23, 25, 39]. While we aimed to consider these best practices when designing the structure of reports, several constraints made this more difficult in our setting. Firstly, we used the original DPWG guideline recommendation texts including their existing data structure, which are tailored to the Dutch setting both in terms of CDS capabilities and in terms of adaptation to the local healthcare system. Secondly, our setting made it necessary to also provide paper-based reports, where the paradigm to provide more information only on request is harder to achieve, as compared to e.g., providing a link to further information in digital CDS tools. Our solution concept was to provide the most relevant information at the top of the report, while including additional information on the following report pages. However, this might still lead to a perceived information overload for some healthcare providers. Improving the design and formatting may, however, help to alleviate this issue.

One rationale of our multi-modal CDS design was to provide a solution for making PGx results and recommendations available and transferable without the need for an EHR infrastructure or the interoperability and interconnectedness between different EHR infrastructures. To achieve this, the Safety Code card was implemented at all sites. The Safety Code card enables report retrieval through standard mobile phones based on a QR-code printed together with a short summary of the PGx profile on a credit-card sized card. While our analysis revealed that the Safety Code card was accepted and considered a beneficial tool among the majority of participating health care providers, one important insight was the fact that this assessment and especially the card's usage seemed to differ between implementation sites. For example, our evaluation showed that the Greek implementation sites, where no EHR infrastructure exists at all, had, overall, relatively lower usage of the card. Similarly, the Spanish and Italian implementation sites showed, compared to other sites and in relation to the number of recruited patients, lower usage of the Safety Code card.

Non-usage may be attributed to low acceptance from patient-side (i.e., patients not carrying their cards or not showing them to healthcare providers), low acceptance from healthcare provider-side or lower number of occasions for card utilization (e.g., lower number of future PGx drug prescriptions). Feedback from clinical coordinators pointed to low acceptance from the healthcare provider-side due to lower digital literacy at the Greek implementation site, and low acceptance from patient-side at the Spanish implementation site. Further, differences in patient population, such as median age and health status might have led to different adoption rates in different countries. For example, patients recruited at the Italian implementation sites were mostly oncological patients with a relatively higher median age. It has to be noted that familiarity with and usage of information and communication technology show large variations across European countries, and differ per age group, which may have also impacted adoption and acceptance [40]. Additionally, we found that harmonization with pre-existing solutions should be given high priority to increase acceptance among healthcare providers.

One inherent limitation of our evaluation survey is potential non-responder bias, i.e., the opinions and views of healthcare providers who chose not to participate in the survey may not be represented adequately. These healthcare providers may have abstained from the survey due to a variety of reasons, e.g., lack of time, unfamiliarity with or low acceptance of web-based surveys or too little experience with the decision support tools. We mitigated this limitation through the following approaches: (1) after three reminders, sending a last email to survey non-respondents asking for free-text feedback or, alternatively, a reason for why they chose to abstain, and (2) including feed-back from clinical coordinators on the acceptance of the tools among the involved healthcare providers.

We used risk matrices to assess potential risks in the pre-implementation phase. While risk matrices have limitations due to their categorical nature and potential oversimplification of risks, they can serve as a valuable tool for getting an overview of potential challenges in a large-scale implementation project when used appropriately [29, 30].

Finally, the statistics on report retrievals via the Safety Code card do not allow us to distinguish between user groups and reasons for scanning the code, e.g., patients vs. healthcare providers, testing the QR code vs. retrieving the report to use it for personalizing drug prescriptions.

## Meta-evaluation & lessons learned

Previous work has pointed out that much PGx implementation work is done in isolation without sharing experiences and best practices [41]. We would therefore like to share insights, experiences and tools collected within this large-scale, multinational implementation project that can be of value to similar implementation projects.

The design and implementation of the U-PGx CDS infrastructure could not have happened without the collaboration of many contributors with diverse backgrounds and expertise, ranging from basic research to clinical PGx to software engineering. Such a multidisciplinary, large-scale and international implementation project, however, also provides challenges in terms of project management and communication.

We used a variety of communication channels throughout the implementation, including regular structured teleconference calls, structured questionnaires and dedicated mailing lists that ensured that emails were received by all relevant persons. One mailing list was established to provide a contact point for clinical sites to report issues or ask questions surrounding the genotyping platform and the CDS tools, while another one was used to enable discussions with, and distribution of information to all relevant persons at clinical sites.

Each teleconference call regarding the technical implementation was transcribed in detail and shared together with a short summary and action items with all collaborators. In the initial phase, we also experimented with using different Gantt chart software tools, such as GanttPRO [42]. While we think that the Gantt chart helped in keeping a broad overview of the different tasks and their interdependencies, we used it less for tracking and updating the more granular tasks, as this would have required every collaborator to be familiar with and actively use the tool.

We chose the online survey platform Alchemer both for conducting an initial risk assessment analysis and for the evaluation of the CDS tools among participating healthcare providers due to its piping and logic features, which helped to keep survey completion time low, and provided the possibility to track response status [31].

Conducting a collaborative and structured assessment of risks was helpful to gain an overview of potential challenges, anticipate issues and circumvent blockers that cannot easily be solved due to e.g. institutional barriers.

## Opportunities and challenges regarding maintenance and extended roll-out in Europe

Within the U-PGx project, we have demonstrated the feasibility and acceptance of a framework for PGx CDS implementation that forms the basis for realizing the vision of harmonized, barrier-free PGx decision support across Europe. Still, maintenance and extended roll-out in Europe will require continued effort and work. This concerns the maintenance and improvement of the CDS framework components including the guidelines, the harmonization of PGx information in drug labels and guidelines, and, ideally, the establishment of standards including genotype-phenotype translations to guarantee interoperability and to secure quality control.

**PGx guidelines.** The DPWG guidelines authored and maintained by one project partner (Royal Dutch Pharmacists Association, KNMP) were used as a source for the PGx recommendations. While these guidelines have been pre-existing and had already been routinely used in primary care and hospital clinic by the Dutch implementation site before the project, resources for maintenance updating and translating guidelines were scaled up as part of the U-PGx project. Since the guidelines form the basis of PGx CDS, the securing of financial and personnel resources are a prerequisite for their further maintenance and potential international expansion.

**Standardisation and harmonisation.** Another essential aspect regarding interoperability and transparency concerns the standardization of PGx variants and phenotypes and the harmonization of information on PGx variants and their clinical relevance. Recent analyses have shown that substantial discrepancies exist between drug labels and guidelines in terms of content and coverage of PGx variants, and also between guidelines authored by different PGx working groups, such as the DPWG and CPIC [43–48].

Continual efforts on standardization and harmonization have been undertaken by guideline-authoring consortia, such as defining a list of consensus terms together with a panel of PGx experts and standardizing genotype to phenotype translation [49, 50]. Furthermore, a standardized minimal panel of PGx variants for pre-emptive PGx testing has recently been proposed by van der Wouden et al. [17].

Consistency of information in drug labels and guidelines is not only important for standardization but may also impact financial coverage by health insurances, and ultimately liability and authorization issues. This has recently been exemplified by the United States Food and Drug Administration's (FDA) warning against making predictive statements about dosing for

gene-drug pairs not on its authorized list which was issued to several providers of PGx tests and CDS solutions [51].

Finally, consistent and actionable PGx information in drug labels might also positively impact physicians' willingness and motivation to consider PGx results in daily practice, as has been pointed out previously [44].

**Software maintenance and compliance with MDR.**   Delivering large-scale PGx decision support for a large number of variants and drugs in a clinical setting requires resources for maintenance, quality control, support and regulatory compliance of software products. In the U-PGx project, these aspects were secured by the bio.logis Genetic Information Management GmbH and bio.logis digital health GmbH who already had experience in delivering PGx CDS solutions including experienced staff and the prerequisites for certification as a medical device under European Medical Device Regulation (MDR).

In Europe, while PGx CDS falls under Regulation (EU) 2017/745 [52], it is, however, not directly regulated and may therefore, depending on the concrete specifications of the system at hand and the type of algorithm used, remain in a legal grey area. While the simple matching of phenotypes with recommendations realized with a lookup table might not constitute a medical device under current legislation, the more complex matching of SNP variants and combinations with phenotypes is more likely to require CE certification when used outside of pure research settings. However, achieving CE mark certification is a time, cost and resource-consuming procedure consisting of several phases, such as obtaining a declaration of conformity and a clinical evaluation report. This may be challenging for research and clinical institutions without dedicated software development departments.

The U-PGx project partner bio.logis digital health GmbH will continue to offer their IT services beyond project end. In addition, it is planned to offer relevant software code in an open source framework. This should enable an interested community to develop further structures, work on improvements and lead to broader distribution. The quality assurance approved elements of the system are in the process of certification as a medical device class 2a. Further, as part of the maintenance strategy, an open source, light-weight version of the Safety Code card system that performs simple matching between phenotypes and recommendations and can be used as a standalone system or integrated with existing infrastructures was developed by the Medical University of Vienna [53, 54].

**Future roll-out scenarios.**   Several future roll-out scenarios that may complement each other could be envisioned. One scenario is the piecewise roll-out of local, commercial solutions as it is already ongoing in many European countries. Advantages of this approach include the coverage of compliance with MDR, support and software maintenance and development is covered by the vendor. Potential problems include system fragmentation and lack of widespread trust among healthcare professionals and patients if no common standard is agreed upon or the standard is not promoted by a recognized institution.

In another scenario, a pan-European organization could take over the role of further maintaining, promoting and distributing the U-PGx decision support tools. This could be a newly created organization, an already existing European-level organization or a large and established company. While this scenario has the clear advantage of the availability of a central institution that promotes and maintains a standardized solution, it is unclear which organization or company could play such a role and how such an institution could be established.

Finally, in a third scenario, a consortium or organization that acts as a standards body could be established, publishing standards and basic solutions and working with local and national organizations. While this scenario, similarly to scenario II, has the advantage of the availability of a central institution that promotes and maintains a set of standards, it entails the challenges of building up including the ongoing acquisition of funding. The establishment of

an international institution to promote global harmonization of PGx information in drug labels and guidelines has also been suggested by Koutsilieri et al. [43].

The proposed scenarios are not necessarily mutually exclusive but could complement each other, e.g., a scenario based on commercial vendors might profit from the establishment of an organization promoting standards for PGx testing and decision support to guarantee quality, transparency and standardization.

## Conclusions

We have demonstrated the feasibility of implementing a standardized PGx decision support solution in a multinational, multi-language and multi-center setting. Overall, evaluation results indicated a satisfactory adoption and acceptance of the provided CDS tool among participating healthcare providers. Further, our evaluation results and process analyses will inform the improvement of the CDS framework to further lower the barrier of clinical PGx implementation in Europe and other regions of the world. Maintenance and extended roll-out across Europe will require ongoing efforts to tackle remaining challenges such as harmonization of existing PGx information in guidelines and drug labels, devising roll-out strategies that lower the barrier for healthcare institutions and providers while ensuring compliance with regulatory and legal frameworks.

## Supporting information

**S1 File. Requirements analysis.** Requirements analysis framework and results.
(PDF)

**S2 File. Regulatory and legal framework analysis.** Analysis of the U-PGx CDS infrastructure in relation to the European regulatory and legal landscape.
(PDF)

**S3 File. Risk assessment.** Questionnaire to identify potential risks and problems of the implementation and results.
(PDF)

**S4 File. CDS evaluation questionnaire.** Questionnaire for evaluating the CDS tools among participating healthcare providers.
(PDF)

**S5 File. STARi checklist.** Filled Standards for Reporting Implementation Studies (StaRI) checklist.
(PDF)

## Acknowledgments

We thank all participating healthcare providers for taking their time to complete the survey.
**Consortium members**
**Consortium lead: Henk-Jan Guchelaar, H.J.Guchelaar [at] lumc.nl**
Kathrin Blagec[1], Jesse J Swen[2], Rudolf Koopmann[3,4], Ka-Chun Cheung[5], Mandy Crommentuijn–van Rhenen[5], Inge Holsappel[5], Lidija Konta[3], Simon Ott[1], Daniela Steinberger[2,4], Hong Xu[1], Erika Cecchin[7], Vita Dolžan[8], Cristina Lucía Dávila Fajardo[9], George P. Patrinos[10], Gere Sunder-Plassmann[11], Richard M Turner[12], Henk-Jan Guchelaar[2], Matthias Samwald[1], M.J. Antolinos-Perez[13], T. Blagus[8], A. Cambon-Thomsen[14], V.H. Deneer[15], M. Dupui[14], P. Franssen[15], L. Grandia[16], A. Hanson[12], M. Ingelman-Sundberg[17], W. Jama[5], S. Jonsson[18], K.

Just[19], M.O. Karlsson[18], T. Katsila[10], J. Klen[8], M. Kriek[2], V.M. Lauschke[16], Xando Díaz-Villamarín[28], L.E.N. Manson[2], C. Mitropoulou[20], G. Mlinsek[21], R. Moreno-Aguilar[29], M. Nijenhuis[5], Å. Nordling[17], M. Pirmohamed[22], A. Poplas Susic[23], E. Rial-Sebbag[14], V. Rollinson[12], J. G. Sanchez-Ramos[28], E. Schaeffeler[24], A. Schmidt[11], M. Schwab[24,25,26], G. Sengoelge[11], M. Steffens[27], J. Stingl[19], G. Toffoli[7], E. Moreno Escobar[28], R. Tremmel[25], E.-E. Tsermpini[10], M. van der Lee[2], J.-G. Wojtyniak[24], C.H. van der Wouden[2]

[1] Institute of Artificial Intelligence and Decision Support; Center for Medical Statistics, Informatics, and Intelligent Systems; Medical University of Vienna, Vienna, Austria.

[2] Department of Clinical Pharmacy & Toxicology, Leiden University Medical Center, Leiden, The Netherlands.

[3] diagnosticum Center for Human Genetics, Frankfurt am Main, Germany.

[4] Institute for Human Genetics, Justus Liebig University, Giessen, Germany.

[5] Medicines Information Centre; Royal Dutch Pharmacists Association (KNMP), The Hague, The Netherlands.

[2] bio.logis digital health GmbH, Frankfurt am Main, Germany.

[7] Experimental and Clinical Pharmacology Unit, Centro di Riferimento Oncologico di Aviano (CRO) IRCCS, Aviano, Italy.

[8] University of Ljubljana, Faculty of Medicine, Institute of Biochemistry and Molecular Genetics, Pharmacogenetics Laboratory, Ljubljana, Slovenia.

[9] Hospital Universitario Virgen de las Nieves, Instituto de Investigación Biosanitaria Granada (Ibs.Granada), Granada, Spain.

[10] University of Patras School of Health Sciences, Department of Pharmacy, Laboratory of Pharmacogenomics and Individualized Therapy, Patras, Greece.

[11] Division of Nephrology and Dialysis, Department of Internal Medicine III, Medical University of Vienna, Vienna, Austria.

[12] The Wolfson Centre for Personalised Medicine; Institute of Systems, Molecular and Integrative Biology; University of Liverpool, Liverpool, UK.

[13] Cardiology Unit, Universitary Hospital Virgen de las Nieves, Granada, Spain.

[14] UMR Inserm U1027 and Université de Toulouse III Paul Sabatier, Toulouse, France.

[15] Department of Clinical Pharmacy, St Antonius Hospital, Nieuwegein, The Netherlands.

[16] Z-Index, The Hague, The Netherlands.

[17] Department of Physiology and Pharmacology, Section of Pharmacogenetics, Karolinska Institutet, Stockholm, Sweden.

[18] Department of Pharmacy, Uppsala University, Box 580, 75123, Uppsala, Sweden.

[19] Institute of Clinical Pharmacology, RWTH Aachen University, Aachen, Germany.

[20] The Golden Helix Foundation, London, United Kingdom.

[21] Department of Nephrology, University Medical Center Ljubljana.

[22] Department of Molecular and Clinical Pharmacology, Royal Liverpool University Hospital and University of Liverpool, Liverpool, United Kingdom.

[23] Ljubljana Community Health Centre, Ljubljana, Slovenia.

[24] Dr. Margarete Fischer-Bosch Institute of Clinical Pharmacology, Stuttgart, Germany and University of Tübingen, Tübingen, Germany.

[25] Department of Clinical Pharmacology, University Hospital Tübingen, Tübingen, Germany.

[26] Department of Pharmacy and Biochemistry, University of Tübingen, Tübingen, Germany.

[27] Research Division, Federal Institute for Drugs and Medical Devices, Bonn, Germany.

[28] Department of Cardiology, Granada University Hospital, Institute for Biomedical Research, ibs., Granada, Spain.

[29] Sub-directorate of Information Technologies and Communications, Hospital Universitario Virgen de las Nieves, Granada

## Author Contributions

**Conceptualization:** Kathrin Blagec, Jesse J. Swen, Munir Pirmohamed, Henk-Jan Guchelaar, Matthias Samwald.

**Data curation:** Rudolf Koopmann, Mandy Crommentuijn - van Rhenen, Inge Holsappel.

**Formal analysis:** Kathrin Blagec.

**Funding acquisition:** Jesse J. Swen, Munir Pirmohamed, Henk-Jan Guchelaar, Matthias Samwald.

**Investigation:** Kathrin Blagec.

**Methodology:** Kathrin Blagec.

**Resources:** Jesse J. Swen, Ka-Chun Cheung, Mandy Crommentuijn - van Rhenen, Inge Holsappel, Lidija Konta, Daniela Steinberger, Erika Cecchin, Vita Dolžan, Cristina Lucía Dávila-Fajardo, George P. Patrinos, Richard M. Turner.

**Software:** Rudolf Koopmann, Lidija Konta, Simon Ott, Daniela Steinberger, Hong Xu.

**Supervision:** Matthias Samwald.

**Visualization:** Kathrin Blagec.

**Writing – original draft:** Kathrin Blagec.

**Writing – review & editing:** Jesse J. Swen, Rudolf Koopmann, Ka-Chun Cheung, Mandy Crommentuijn - van Rhenen, Inge Holsappel, Lidija Konta, Daniela Steinberger, Erika Cecchin, Vita Dolžan, Cristina Lucía Dávila-Fajardo, George P. Patrinos, Gere Sunder-Plassmann, Richard M. Turner, Henk-Jan Guchelaar, Matthias Samwald.

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
