## [Decision Letter · Decision Letter 0]

9 Nov 2021

PONE-D-21-24987

Pharmacogenomics decision support in the U-PGx project: Results and advice from clinical implementation across seven European countries

PLOS ONE

Dear Dr. Samwald,

Thank you for submitting your manuscript to PLOS ONE. After careful consideration, we feel that it has merit but does not fully meet PLOS ONE’s publication criteria as it currently stands. Therefore, we invite you to submit a revised version of the manuscript that addresses the points raised during the review process.

Thank you for your patience during the peer review period, it was longer than expected, since it was challenging to invite appropriate peer reviewers for your manuscript. Besides considering the reviewer comments at your revision, I kindly ask you to include the name of the clinical trial registry and registration number in the Abstract. 

We look forward to receiving your revised manuscript.

Kind regards,

János G. Pitter, MD, PhD

Academic Editor

PLOS ONE

2. During our internal checks, the in-house editorial staff noted that you conducted research or obtained samples in another country. Please check the relevant national regulations and laws applying to foreign researchers and state whether you obtained the required permits and approvals. Please address this in your ethics statement in both the manuscript and submission information.

“DS has developed concepts of genetic information management (GIM) that are realized by the company bio.logis digital health GmbH. For the company, she is CEO. She is also medical director of diagnosticum Center of Human Genetics, a diagnostic institution that is applying and testing the GIM-systems in the context of medical care. GPP is Full Member and National representative of the European Medicines Agency, Committee for Human Medicinal Products (CHMP) - Pharmacogenomics Working Party, Amsterdam, the Netherlands. The other authors have no competing interests to declare.”

4. One of the noted authors is a group or consortium Ubiquitous Pharmacogenomics Consortium. In addition to naming the author group, please list the individual authors and affiliations within this group in the acknowledgments section of your manuscript. Please also indicate clearly a lead author for this group along with a contact email address.

Additional Editor Comments:

Please include the name of the clinical trial registry and registration number in the Abstract. 

Reviewers' comments:

Reviewer's Responses to Questions

**Comments to the Author**

1. Is the manuscript technically sound, and do the data support the conclusions?

Reviewer #1: Yes

Reviewer #2: Yes

2. Has the statistical analysis been performed appropriately and rigorously? 

Reviewer #1: Yes

Reviewer #2: Yes

3. Have the authors made all data underlying the findings in their manuscript fully available?

Reviewer #1: Yes

Reviewer #2: Yes

4. Is the manuscript presented in an intelligible fashion and written in standard English?

Reviewer #1: Yes

Reviewer #2: Yes

5. Review Comments to the Author

Reviewer #1: From a randomized crossover study design, the authors report results of clinical decision support systems (CDS) implementation in the large-scale European research project Ubiquitous Pharmacogenomics (U-PGx), in which PGx CDS was rolled out and evaluated across more than 15 clinical sites. Personalized pharmacogenomics (PGx) reports were generated from 6884 genotyped samples. Participating healthcare providers expressed overall satisfactory adoption and acceptance of the CDS tools.

Minor revisions:

1- In addition to stating counts, provide corresponding percentages on the following pages: 11, 15-16, 17.

2- Line 343 indicates that averages were reported in table 3 but the column is labelled medians. Clarify.

3- Cite the statistical software used for the analysis.

4- Figure 5: Include the percentage of respondents in each country.

5- Figures 6-11: Include percentages which correspond to the counts.

Reviewer #2: Clinical Decision Support is a potentially crucial tool for widespread adoption of pharmacogenomics in clinical practice. In this manuscript, the authors analyze the rollout of a PGx CDS implementation project that spans multiple sites and countries in Europe. Strengths of the study include its multi-site, international nature and its mixed-methods approach for data collection. This is a potentially valuable study and the data collection, analysis, and results are generally well done. However, there are a number of issues with the paper's overall organization and flow, which should be corrected. Additionally, the Discussion section does not address some of the more interesting results that I expect readers could learn the most from. Feedback follows, organized by section:

Abstract

• The methods section references analysis of "workflow bottlenecks," but this terminology isn't used in the methods and results section, so it's not clear to me what this is specifically referring to.

• The majority of the Results section of the abstract reflects commentary from the Discussion section, not the results of the systematic analysis. I suggest reorganizing to make this clear to the reader and instead emphasizing the reported results.

Background

• Context should be expanded. A number of other PGx CDS programs have reported outcomes of their projects, so what are the specific literature gaps/unknowns this study addresses? U-PGx's international, multi-site nature is unique, so several can be inferred, but the authors should more explicitly state what they set out to learn with this study that's different from other PGx CDS implementation retrospectives.

Methods

• The Methods section contains editorial-type statements that are better suited to the Background or Discussion sections, including:

o A statement starting on line 129 providing a definition for "pre-emptive PGx testing"

o A statement starting on line 143 providing commentary on the necessity of PGx CDS

o A statement starting on line 217 regarding the value of risk matrices

• The Evaluation framework section lists four methods for data collection in this study. The following sub-sections explicitly define methods 2-4, but the first method, "tracking of general process indicators" is not explicitly defined. It's not clear to me what this means, what is being measured, or how the data was collected. (Additionally, the "Statistics on report retrievals" heading appears to be incorrectly formatted as a higher-level heading, but that could be an artifact of my reader – I suggest double checking the formatting to be sure.)

Results

• The "General implementation process indicators" section begins with several statements on methodology that are more appropriate for a Methods section. Reorganizing this would also help address my comment above.

• The Demographics section under "Decision support infrastructure evaluation survey" reports the demographics of those who responded to the survey, but it's not clear how representative these respondents are of overall U-PGx users. Are any statistics available on basic user demographics to put these numbers into context? For example, does a 30.8% proportion accurately reflect the number of Pharmacist users, or are they over/under-represented?

Discussion

• The statement starting on line 596 mis-characterizes the FDA's stance on PGx testing. The FDA has not banned PGx testing or CDS for gene-drug pairs that are not on their list. Rather, they have warned sellers of PGx tests and PGx CDS against making predictive statements about drug dosing that have not been verified by the FDA. Labs are still able to test PGx-related genes and CDS tools for a variety of drug-gene pairs are still available, but their predictive statements are limited. The following URL should be referenced, in addition to the tables already referenced: https://www.fda.gov/medical-devices/precision-medicine/table-pharmacogenetic-associations

• There were several interesting positive findings in the Results section that were not directly addressed in the Discussion, which I would like to see further comment on, including:

o Users reported a high level of satisfaction with the PGx training they received, which is in contrast with typical reports showing low familiarity with PGx among providers. What is it about U-PGx that led to this level of satisfaction?

o Similarly, users reported high satisfaction for workflow fit, information needs being met, and system user-friendliness. Previous projects have struggled in these areas. What did U-PGx do that made this successful?

6. PLOS authors have the option to publish the peer review history of their article (what does this mean?). If published, this will include your full peer review and any attached files.

Reviewer #1: No

Reviewer #2: No

---

## [Author Response · Author response to Decision Letter 0]

14 Jan 2022

We thank the reviewers and editor for helping us improve our manuscript! Replies to comments are listed below.

Editor comments:

“1. Please ensure that your manuscript meets PLOS ONE's style requirements, including those for file naming. The PLOS ONE style templates can be found at

https://journals.plos.org/plosone/s/file?id=wjVg/PLOSOne_formatting_sample_main_body.pdf and https://journals.plos.org/plosone/s/file?id=ba62/PLOSOne_formatting_sample_title_authors_affiliations.pdf”

→ We revised the formatting of the manuscript including the file names to comply with PLOS ONE’s requirements.

“2. During our internal checks, the in-house editorial staff noted that you conducted research or obtained samples in another country. Please check the relevant national regulations and laws applying to foreign researchers and state whether you obtained the required permits and approvals. Please address this in your ethics statement in both the manuscript and submission information.”

→ The study was approved by ethics review boards in all participating countries and all necessary permits and approvals were obtained. We added this information to the ethics statements.

We have now also included the names of ethics review boards in all participating countries.

“3. Thank you for stating the following in the Competing Interests section:

“DS has developed concepts of genetic information management (GIM) that are realized by the company bio.logis digital health GmbH. For the company, she is CEO. She is also medical director of diagnosticum Center of Human Genetics, a diagnostic institution that is applying and testing the GIM-systems in the context of medical care. GPP is Full Member and National representative of the European Medicines Agency, Committee for Human Medicinal Products (CHMP) - Pharmacogenomics Working Party, Amsterdam, the Netherlands. The other authors have no competing interests to declare.”

Please include your updated Competing Interests statement in your cover letter; we will change the online submission form on your behalf. “

→ We included an updated competing interests statement in the cover letter.

“4. One of the noted authors is a group or consortium Ubiquitous Pharmacogenomics Consortium. In addition to naming the author group, please list the individual authors and affiliations within this group in the acknowledgments section of your manuscript. Please also indicate clearly a lead author for this group along with a contact email address.”

→ We added the requested information to the acknowledgements section.

“5. Your ethics statement should only appear in the Methods section of your manuscript. If your ethics statement is written in any section besides the Methods, please delete it from any other section.”

→ We moved the ethics statement to the Methods section of the manuscript.

Reviewer #1: 

“From a randomized crossover study design, the authors report results of clinical decision support systems (CDS) implementation in the large-scale European research project Ubiquitous Pharmacogenomics (U-PGx), in which PGx CDS was rolled out and evaluated across more than 15 clinical sites. Personalized pharmacogenomics (PGx) reports were generated from 6884 genotyped samples. Participating healthcare providers expressed overall satisfactory adoption and acceptance of the CDS tools.

Minor revisions:

1- In addition to stating counts, provide corresponding percentages on the following pages: 11, 15-16, 17.”

→ We added the percentages in the main text.

“2- Line 343 indicates that averages were reported in table 3 but the column is labelled medians. Clarify.”

→ We corrected this by replacing ‘average’ with ‘median’ in the text.

“3- Cite the statistical software used for the analysis.”

→ We added information on the software used for conducting the analyses in the ‘Methods’ section.

“4- Figure 5: Include the percentage of respondents in each country.”

→ We extended Figure 5 to show the percentage of respondents per country.

“5- Figures 6-11: Include percentages which correspond to the counts.”

→ We updated all figures to include the percentages in addition to the counts.

Reviewer #2: 

“Clinical Decision Support is a potentially crucial tool for widespread adoption of pharmacogenomics in clinical practice. In this manuscript, the authors analyze the rollout of a PGx CDS implementation project that spans multiple sites and countries in Europe. Strengths of the study include its multi-site, international nature and its mixed-methods approach for data collection. This is a potentially valuable study and the data collection, analysis, and results are generally well done. However, there are a number of issues with the paper's overall organization and flow, which should be corrected. Additionally, the Discussion section does not address some of the more interesting results that I expect readers could learn the most from. Feedback follows, organized by section:

Abstract

• The methods section references analysis of "workflow bottlenecks," but this terminology isn't used in the methods and results section, so it's not clear to me what this is specifically referring to.”

→ We added an explanation for this term in the methods section on page 12.

“• The majority of the Results section of the abstract reflects commentary from the Discussion section, not the results of the systematic analysis. I suggest reorganizing to make this clear to the reader and instead emphasizing the reported results.”

→ We re-structured and extended the abstract to highlight the results of our analyses.

“Background

• Context should be expanded. A number of other PGx CDS programs have reported outcomes of their projects, so what are the specific literature gaps/unknowns this study addresses? U-PGx's international, multi-site nature is unique, so several can be inferred, but the authors should more explicitly state what they set out to learn with this study that's different from other PGx CDS implementation retrospectives.”

→ We extended the ‘Background’ section to more explicitly describe how our study differentiates from existing CDS implementation projects and what our study adds to the existing literature. Indeed we especially highlight the multi-national, multi-site, multi-lingual and multi-modal nature of our intervention and study.

“Methods

• The Methods section contains editorial-type statements that are better suited to the Background or Discussion sections, including:

o A statement starting on line 129 providing a definition for "pre-emptive PGx testing"”

→ We moved the section describing the setting of the study including the definition of preemptive PGx testing to the ‘Background’ section. 

“o A statement starting on line 143 providing commentary on the necessity of PGx CDS”

→ We removed the corresponding sentence.

“o A statement starting on line 217 regarding the value of risk matrices”

→ We moved the statement on risk matrices to the ‘Discussion’ section. 

“• The Evaluation framework section lists four methods for data collection in this study. The following sub-sections explicitly define methods 2-4, but the first method, "tracking of general process indicators" is not explicitly defined. It's not clear to me what this means, what is being measured, or how the data was collected.”

→ We added a paragraph to the ‘Methods’ section that describes the process indicators and how these data were collected.

“(Additionally, the "Statistics on report retrievals" heading appears to be incorrectly formatted as a higher-level heading, but that could be an artifact of my reader – I suggest double checking the formatting to be sure.)”

→ Indeed it was incorrectly formatted; we corrected the formatting of the heading.

“Results

• The "General implementation process indicators" section begins with several statements on methodology that are more appropriate for a Methods section. Reorganizing this would also help address my comment above.”

→ We moved these explanatory statements to the newly added paragraph on ‘process indicators’ in the ‘Methods’ section (see our reply above).

“• The Demographics section under "Decision support infrastructure evaluation survey" reports the demographics of those who responded to the survey, but it's not clear how representative these respondents are of overall U-PGx users. Are any statistics available on basic user demographics to put these numbers into context? For example, does a 30.8% proportion accurately reflect the number of Pharmacist users, or are they over/under-represented?”

→ We added information on demographic characteristics of the overall invited healthcare providers that were available to us in the ‘Demographics’ section to make it clear how representative our sample is for the total number of users. The gender ratio of respondents matches that of invited participants. Based on the institutions where invited participants were employed, we infer that pharmacists were somewhat over-represented and primary care physicians were somewhat under-represented among respondents. However, these fluctuations do not significantly impact our overall findings, since all groups were sufficiently covered by the pool of respondents.

“Discussion

• The statement starting on line 596 mis-characterizes the FDA's stance on PGx testing. The FDA has not banned PGx testing or CDS for gene-drug pairs that are not on their list. Rather, they have warned sellers of PGx tests and PGx CDS against making predictive statements about drug dosing that have not been verified by the FDA. Labs are still able to test PGx-related genes and CDS tools for a variety of drug-gene pairs are still available, but their predictive statements are limited. The following URL should be referenced, in addition to the tables already referenced: https://www.fda.gov/medical-devices/precision-medicine/table-pharmacogenetic-associations”

→ Thanks for pointing this out! We adapted the text to more accurately reflect the FDA’s stance. We noticed that the link referenced in this comment is the same as the one already cited in the references.

“• There were several interesting positive findings in the Results section that were not directly addressed in the Discussion, which I would like to see further comment on, including:

o Users reported a high level of satisfaction with the PGx training they received, which is in contrast with typical reports showing low familiarity with PGx among providers. What is it about U-PGx that led to this level of satisfaction?”

→ We added a paragraph in the discussion section that provides possible explanations for the high level of satisfaction with PGx CDS training.

“o Similarly, users reported high satisfaction for workflow fit, information needs being met, and system user-friendliness. Previous projects have struggled in these areas. What did U-PGx do that made this successful?”

→ We extended the discussion section to provide potential explanations for these positive findings compared to previous PGx CDS implementation projects.

---

## [Editor Report · Decision Letter 1]

3 May 2022

Pharmacogenomics decision support in the U-PGx project: Results and advice from clinical implementation across seven European countries

PONE-D-21-24987R1

Dear Dr. Samwald,

We’re pleased to inform you that your manuscript has been judged scientifically suitable for publication and will be formally accepted for publication once it meets all outstanding technical requirements.

Kind regards,

János G. Pitter, MD, PhD

Academic Editor

PLOS ONE

Additional Editor Comments (optional):

Thank you for your patience in the peer review period, it took longer than expected to secure the necessary peer reviews.   
---

## [Editor Report · Acceptance letter]

26 May 2022

PONE-D-21-24987R1 

Pharmacogenomics decision support in the U-PGx project: Results and advice from clinical implementation across seven European countries 

Dear Dr. Samwald:

I'm pleased to inform you that your manuscript has been deemed suitable for publication in PLOS ONE. Congratulations! Your manuscript is now with our production department. 

Kind regards, 

on behalf of

Dr. János G. Pitter 

Academic Editor

PLOS ONE